# Subset of DN Memory B Cells Expressing Low Levels of Inhibitory Receptor BTLA Is Enriched in SLE Patients

**DOI:** 10.3390/cells13242063

**Published:** 2024-12-13

**Authors:** Lucie Aubergeon, Renaud Felten, Jacques-Eric Gottenberg, Hélène Dumortier, Fanny Monneaux

**Affiliations:** 1Immunology, Immunopathology and Therapeutic Chemistry, Institute of Molecular and Cellular Biology, CNRS UPR3572, 67084 Strasbourg, France; 2Rheumatology Department, National Reference Center for Autoimmune Diseases, Strasbourg University Hospital, 67000 Strasbourg, France

**Keywords:** systemic lupus erythematosus, BTLA, DN memory B cells, inhibitory receptors

## Abstract

The dialogue between T and B cells can be regulated by different mechanisms, such as co-inhibitory receptors, which therefore play a crucial role in preventing autoimmune diseases such as systemic lupus erythematosus (SLE). B and T lymphocyte attenuator (BTLA) is a co-inhibitory receptor expressed on many myeloid and lymphoid cells. Although peripheral B cells express a very high amount of BTLA, previous works in the context of autoimmunity mainly focused on T cells, and whether BTLA expression on B cells plays a role in the lupus pathogenesis is still unclear. In the present study, we examine the expression of BTLA, as well as its ligand HVEM (Herpesvirus Entry Mediator), on various B cell subsets in lupus patients compared to healthy controls (HCs). We evidenced the existence of double-negative (DN; IgD^−^CD27^−^) memory B cells expressing very low levels of BTLA, which are enhanced in active lupus patients. An in-depth analysis revealed that these BTLA^low^ DN cells mainly correspond to the newly reported DN3 B cell subset, originally described in the context of SARS-CoV2 infection. These cells display an activated and antibody-secreting cell phenotype, and we propose that their low BTLA expression may favor their expansion and rapid differentiation into plasmablasts in lupus patients.

## 1. Introduction

B and T lymphocyte attenuator (BTLA) is an inhibitory receptor of the CD28 superfamily and is expressed at various levels by almost all immune cells [1,2]. BTLA mediates its regulatory functions upon binding its ligand HVEM (Herpesvirus Entry Mediator) [3], a member of the TNF (Tumor Necrosis Factor) receptor family, which is widely expressed on hematopoietic and non-hematopoietic cells [4]. HVEM has the unique ability to bind multiple ligands, such as lymphotoxin-α, CD160, and LIGHT (homologous to lymphotoxin, exhibits inducible expression and competes with HSV (Herpes Simplex Virus) glycoprotein D for binding to herpesvirus entry mediator, a receptor expressed on T lymphocytes). Together, HVEM and its ligands form a multidirectional signaling axis. Indeed, the binding of BTLA or CD160 to HVEM induces an inhibitory signal in cells expressing these receptors, whereas LIGHT and lymphotoxin-α generate a positive activation signal. Additionally, the binding of BTLA to HVEM triggers proliferation and survival signals within the HVEM-expressing cells. Concerning BTLA signaling, its inhibitory function was demonstrated by in vitro hyperproliferative responses of BTLA-deficient B and T cells to specific BCR and TCR (B and T Cell Receptor) stimulation [3]. Additionally, BTLA plays an important role in maintaining peripheral tolerance, as BTLA-deficient mice develop autoimmune symptoms [3,5]. 

Systemic lupus erythematosus (SLE) is a multisystem autoimmune disease characterized by the production of autoantibodies, which contribute to tissue inflammation by depositing as immune complexes in target organs such as skin or kidneys [6]. T–B crosstalk, leading to B cell differentiation into autoantibody-secreting cells, thus plays a crucial role in lupus pathogenesis. In healthy individuals, the T–B interaction is negatively regulated in order to avoid excessive B and T cell activation and unwanted self-specific responses. Several mechanisms are involved in this process, including cellular actors such as regulatory T cells (Tregs), and molecular players such as co-inhibitory receptors. Although the contribution of defective Treg frequencies and/or function has been largely studied [7,8,9,10], there are relatively few studies that have focused on the role of BTLA in SLE [11,12]. In the MRL^lpr/lpr^ lupus mouse model, the genetic deletion of BTLA leads to the development of exaggerated lymphoproliferation, infiltration of inflammatory cells in organs, and an increase in autoantibody production. This enhancement of lupus symptoms is associated with a shortened survival rate [13], suggesting a protective role of the BTLA pathway in lupus pathogenesis. In SLE patients, our group reported an enhanced BTLA expression on activated Tregs (but not resting Tregs) in patients with an active disease, correlating with the reduced frequency of this subset [14]. Moreover, we showed that the induction of the BTLA pathway does not properly suppress CD4^+^ T cell activation and proliferation in lupus patients compared to healthy controls (HCs) [15]. This altered BTLA functionality is due to a poor BTLA recruitment to TCR clusters following activation and can be restored by normalizing lipid metabolism. Although peripheral B cells express very high BTLA levels [16], previous work concerning BTLA expression in the context of autoimmunity mainly focused on T cells, and whether BTLA expression on B cells plays a role in lupus pathogenesis is still unclear. Wiedemann et al. reported a reduced BTLA expression on naive (CD27^−^IgD^+^) and double-negative (DN; CD27^−^IgD^−^) memory B cells from lupus patients compared to HCs [17]. BTLA expression (on naive but not DN B cells) inversely correlated with SICLEC-1 (Sialic Acid-Binding Immunoglobulin-Type Lectins; CD169) expression on monocytes and with anti-dsDNA antibody titers but was not associated with disease activity. However, Vendel et al. recently compared BTLA and HVEM expression using mass cytometry by time-of-flight between HCs and five SLE patients during a flare. The authors did not notice differential BTLA expression on B cells or B cell subsets, whereas HVEM protein levels were found to be significantly decreased in B cells (and plasmacytoid dendritic cells) in patients with SLE [18].

In the present study, we examine the expression of BTLA, as well as that of its ligand HVEM, on various B cell subsets in lupus patients compared to HCs. We very interestingly evidenced the existence of a DN memory B cell subset expressing very low levels of BTLA, which is enhanced in active lupus patients. 

## 2. Materials and Methods

### 2.1. SLE Patients and Healthy Subjects

Forty-eight SLE patients at University Hospital (Strasbourg, France) were enrolled in this study. All patients met the American College of Rheumatology criteria for classification of SLE [19], and disease activity was assessed by the SLE disease activity index (SLEDAI). Anti-dsDNA levels were screened by ELISA (Kallestad anti-DNA microplate EIA, Bio-Rad Lab. Inc., Hercules, CA, USA). The clinical and biological characteristics of SLE patients are listed in Table 1. Patients who received immunosuppressive agents or biologics were excluded from the study, and the included patients were either untreated or treated with hydroxychloroquine, methotrexate, and/or a dose of steroids less than 15 mg per day. HC samples (*n* = 32, age range from 21 to 63 years, median of 41.2) were obtained from the French National Blood Bank (Etablissement Français du sang). 

### 2.2. Ethics Statement

This study is in accordance with French and European ethics standards and was approved by an independent ethical committee (number 2023-A00612-43). All subjects signed an informed consent form, according to the Helsinki declaration. 

### 2.3. Flow Cytometry Analysis

Human peripheral blood mononuclear cells (PBMCs) were isolated from fresh peripheral blood by density gradient (Histopaque-1077; Saint Louis, MO, USA, Sigma-Aldrich) and stained for 20 min at 4 °C in PBS-2%FCS followed by fluorescent-labeled antibodies: IgD-FITC (clone IA6-2; 1/10), CD27-APC (cloneM-T271; 1/10), CD38-PE-Cy7 (clone HIT2; 1/20) from BD Pharmingen (San Diego, CA, USA), BTLA-PE (clone MIH26; 1/20), HVEM-PE-Cy7 (clone 122; 1/20), SLAMF7-PE-Dazzle 594 (clone 162.1; 1/20) from Biolegend (San Diego, CA, USA), CD27-APC-Vio770 (REA499; 1/50), CD19-VioBlue (clone LT19; 1/100), CD95-APC-Vio770 (cloneDX2; 1/10), CXCR5-PE-Vio615 (REA103; 1/50), CD11c-APC-Vio770 (REA618; 1/50), IgG-PerCP-Vio770 (clone ISAA-3B2.2.3; 1/10) from Miltenyi (Bergisch Glabdash, Germany), and HLA-DR-AF700 (clone L203; 1/20) from R&D (Abingdon, UK). To assess T-bet expression, cells were fixed in True-Nuclear 1X Fix Concentrate (Biolegend) for 45 min. Cells were then stained with anti-T-bet-PE-Cy7 (clone 4B10; 1/50; Biolegend) in True-Nuclear 1× Perm Buffer (Biolegend). Single cells were discriminated from aggregates or doublets using SS-W vs. SS-H and FS-W vs. FS-H plots. Cell acquisition was performed using a 10-color Flow Cytometer Gallios-Navios (Beckman Coulter, Brea, CA, USA). At least 1 × 10^6^ cells were analyzed using FlowJo 7.6.5 or 10 software (TreeStar) with the strategy depicted in Figure 1A and Appendix A using Fluorescence Minus One (FMO) controls were used to define the gates.

### 2.4. Statistical Analyses

Data were analyzed using GraphPad Prism version 6 or 8 (GraphPad Software). Differences between HC and SLE patients were determined with a Mann–Whitney test. Relationships between two variables were evaluated using Spearman’s correlation coefficient. The significance of differences between groups was determined using a Kruskal–Wallis test with Dunn’s multiple comparisons. *p*-values < 0.05 were considered significant.

## 3. Results

### 3.1. BTLA and HVEM Expression by Human Lupus B Cells

The expression of BTLA and HVEM on B cell subsets was analyzed by multi-color cytometry to identify, among CD19^+^ B cells, mature naive B cells (IgD^+^CD27^−^), switched memory (SM; IgD^−^CD27^+^) B cells, non-switched memory (NSM; IgD^+^CD27^+^) B cells, DN memory B cells (IgD^−^CD27^−^), and plasmablasts (PBs; CD27^hi^CD38^hi^) (Appendix A and Figure 1A). We confirmed the published abnormalities [20,21,22,23] in the distribution of these B cell subsets in SLE patients compared to partially age-matched HCs, i.e., decreased frequency of NSM B cells (18.7% ± 1 in HCs vs. 10.2% ± 1.3 in SLE patients; *p* < 0.0001), elevated frequency of SM B cells (19.6% ± 1.3 in HCs vs. 26% ± 2 in SLE patients; *p* < 0.05) and of DN memory B cells (4.7% ± 0.3 in HCs vs. 10.7% ± 1.3 in SLE patients; *p* < 0.0001), and a tendency toward a higher percentage of PBs (1.6% ± 0.3 in HCs vs. 2.9% ± 0.5 in SLE patients; ns; *p* = 0.054) (Figure 1B). The analysis of BTLA expression on these B cell subsets revealed that although BTLA is present at the surface of all B cell subsets, its expression is significantly higher on naive IgD^+^ B cells compared to SM and DN IgD^−^ memory B cells both in HC and SLE patients (Appendix A). We also confirmed that contrary to other inhibitory receptors, BTLA is also expressed on plasmablasts. When we compared BTLA levels between HC and SLE patients, we found a decrease in BTLA expression in lupus DN memory B cells (MFI: 24,730 ± 1763 in HCs vs. 17,950 ± 1215 in SLE patients; *p* < 0.01) (Figure 1C). HVEM expression was found to be slightly but significantly lower in SM B cells compared to NSM B cells in HCs (Appendix A) but not SLE patients and was similarly expressed by all B cell subsets in SLE patients compared to HCs (Figure 1D). 

**Figure 1 cells-13-02063-f001:**
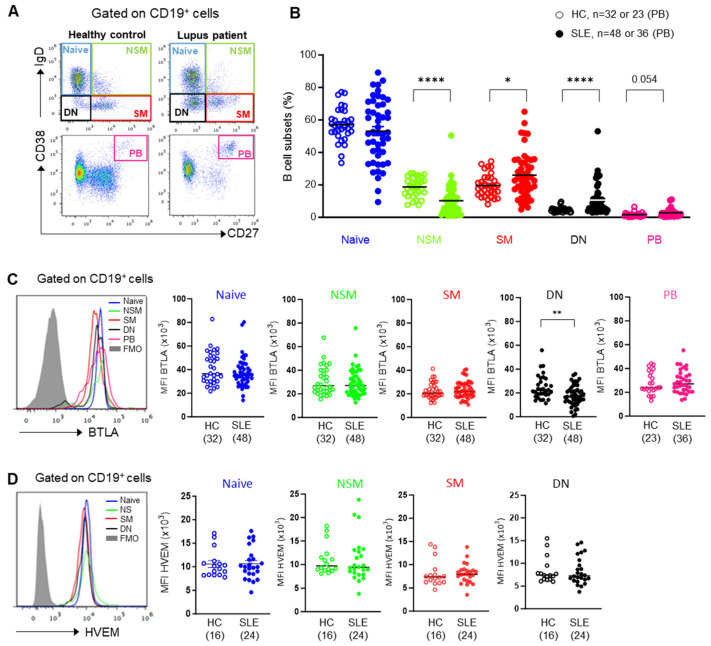
BTLA and HVEM expression on B cells. (**A**) B cell subsets were identified by cytometry among B cells (CD19^+^) as naive B cells (CD19^+^IgD^+^CD27^−^), non-switched memory B cells (NSM; CD19^+^IgD^+^CD27^+^), switched memory B cells (SM; CD19^+^IgD^−^CD27^+^), double-negative memory B cells (DN; CD19^+^IgD^−^CD27^−^), and plasmablasts (PB; CD19^+^CD27^hi^CD38^hi^). (**B**) Frequencies of B cell subsets in HC and SLE patients are shown. (**C**,**D**) Comparison of BTLA (**C**) and HVEM (**D**) expression on B cell subsets in HC and SLE patients. Results are expressed as MFI, and horizontal lines represent the mean of BTLA or HVEM expression. * *p* < 0.05; ** *p* < 0.01; **** *p* < 0.0001, Kruskal–Wallis or Mann–Whitney tests.

### 3.2. BTLA^low^ DN Memory B Cells Are Enhanced in Lupus Patients

BTLA expression was found to be lower on DN memory B cells from SLE patients and interestingly, the frequency of this B cell population is higher in SLE patients than in HCs and particularly in active SLE patients (4.7% ± 0.3 in HCs vs. 8.7% ± 1.7 in patients with inactive SLE or with low disease activity, SLEDAI < 6; *p* < 0.01; and vs. 13.6% ± 2.1 in patients with mild to severe SLE, SLEDAI ≥ 6; *p* < 0.0001) (Figure 2A). Reduced BTLA expression on DN B cells did not seem to be related to the treatments (no treatment, treatment with or without methotrexate) received by SLE patients and did not correlate with age (Figure 2B,C). However, when patients were classified according to their age (< or >60 years old), we noticed a decrease in BTLA levels in DN from lupus patients older than 60 years of age (MFI: 12,330 ± 2010 for patients > 60 years old vs. 19,820 ± 1440 for patients < 60 years old; *p* < 0.05) (Figure 2D). To clarify whether decreased BTLA expression on DN cells is related to age or the disease, we focused on BTLA levels in SLE patients younger than 60 years of age, and we still noticed a reduced BTLA expression compared to HCs (*p* < 0.05; Figure 2E).

Moreover, we showed that among DN memory B cells (but not other B cell subsets), there is a cell population expressing very low levels of BTLA (Figure 3A). This BTLA^low^ population can be detected among DN B cells from HCs (Appendix A), but its frequency is significantly enhanced in lupus (7.4% ± 1 in HCs vs. 18.3% ± 2.7 in SLE patients; *p* < 0.01) (Figure 3B), particularly in patients harboring anti-dsDNA antibodies (*p* < 0.01; Figure 3C). The reduced BTLA expression on DN B cells logically inversely correlates with BTLA^low^ DN B cell frequency, suggesting that BTLA downregulation could favor B cell proliferation and increased numbers of this particular B cell subset in lupus settings (*p* < 0.0001; Figure 3D). The expansion of BTLA^low^ DN B cells also positively correlates with age in SLE patients (Figure 3E; *p* = 0.016). Moreover, BTLA^low^ DN B cells express significantly lower levels of HVEM than BTLA^+^ DN B cells (MFI: 3962 ± 388 for BTLA^low^ vs. 7982 ± 241 for BTLA^+^; *p* < 0.0001) but higher levels of CD95 (MFI: 3554 ± 412 for BTLA^low^ vs. 1710 ± 216 for BTLA^+^; *p* < 0.0001), both in lupus patients (Figure 3F,G) and HCs (Appendix A), which is known to be induced upon B cell stimulation [24], indicating a more activated phenotype of these cells. 

### 3.3. BTLA^low^ DN B Cells Display an Antibody-Secreting Cell Phenotype

We next performed a detailed analysis of these BTLA^low^ DN B cells in order to decipher whether this subset could be associated with antibody-secreting cell functions. As shown in Figure 4A, BTLA^low^ DN B cells express low levels of membrane IgG compared to BTLA^+^ DN B cells (*p* < 0.05). However, the intracellular staining of IgG, allowing the detection of total IgG levels (membrane + intracellular), revealed that BTLA^low^ DN B cells express high levels of total IgG (11.4% ± 2.3 for membrane IgG vs. 49.4% ± 9 for total IgG; *p* < 0.001) (Figure 4A and Appendix A), indicating that these cells can produce IgG that are stored in cytoplasmic vesicles. 

These cells also display a size higher than BTLA^+^ DN B cells, as shown by the FSC (forward scatter) parameter, but smaller than plasmablasts (Figure 4B). Moreover, compared to BTLA^+^ DN B cells, BTLA^low^ DN B cells from SLE patients express higher levels of CD38 (MFI: 16,470 ± 3652 vs. 6827 ± 806 for BTLA^+^ cells; *p* < 0.05) and SLAMF7 (SLAM Family Member 7) (MFI: 2986 ± 385 vs. 1357 ± 111 for BTLA^+^ cells; *p* < 0.001)—which are classical markers for plasma cells—and do not express HLA-DR, which is also absent from the plasma cell surface (MFI: 20,740 ± 1395 for BTLA^+^ B cells vs. 1731 ± 475 for BTLA^low^ B cells; *p* <0.001) (Figure 4C). This in-depth comparison was not performed in all HC samples (3 to 10 HC samples were analyzed); however, it seems that BTLA^low^ DN B cells from HCs exhibit a phenotype similar to SLE-derived BTLA^low^ DN B cells (Appendix A). Altogether, our data provide evidence that BTLA^low^ DN B cells share several characteristics with antibody-secreting cells.

### 3.4. BTLA^low^ DN B Cells Mainly Correspond to the Newly Described DN3 B Cell Subset

Among DN B cells, a subset expressing the integrin CD11c and the transcription factor T-bet was described to be expanded in SLE patients [25]. These CD11c^+^Tbet^+^ DN B cells, named atypical B cells, are prone to differentiate into autoantibody-secreting cells and are present in nephrotic kidneys, thus strongly suggesting that they contribute to lupus pathogenesis. We next compared the frequency of atypical B cells in HC and SLE patients. We observed a trend toward an increase in CD11c^+^Tbet^+^ DN B cell percentages in SLE patients even when significance was not reached (Figure 5A), probably due to small sample size. However, lupus CD11c^+^Tbet^+^ DN memory B cells do not seem to display altered BTLA expression compared to HCs (Figure 5B).

Other recent studies showed that the DN memory population is heterogeneous and can be subdivided into 4 subsets based on differential expression of CD11c and CD21/CXCR5 [26,27]. We thus analyzed whether BTLA^low^ DN B cells could be related to one of these DN subsets. We observed that BTLA^low^ DN B cells lack CXCR5 (like DN2), express low levels of CD19 (like DN1), and do not uniformly express CD11c (Figure 5C). The BTLA^low^ DN B cells we described thus resemble the DN3 subset. Accordingly, when BTLA expression was analyzed on gated DN subsets, we confirmed that the lowest BTLA expression was observed on DN3 B cells (Figure 5D,E; *p* < 0.0001), although DN2 B cells express lower levels of BTLA than DN1 B cells (*p* < 0.05) in SLE patients but also in HCs (Appendix A). Inversely, DN3 B cells represent the main subset among BTLA^low^ DN B cells (61 ± 20% for DN3 vs. 26 ± 19% for DN2 or 7 ± 5% for DN1 B cells; Figure 5F).

## 4. Discussion

Dysregulation of B cell homeostasis is a cardinal feature of SLE. Several B-cell-related defects, including hyperactivity, hypergammaglobulinemia, and spontaneous autoantibody production in vitro [28] have been widely reported in SLE. Alterations in B cell subsets also play a critical role in the pathogenesis of the disease. These changes are characterized by the contraction of naive and NSM B cells [20] along with the expansion of some B cell subsets, such as plasmablasts [21], transitional B cells [29], SM B cells, and DN memory B cells [23]. In our SLE cohort, we observed elevated percentages of IgD^−^ memory B cells (both SM and DN B cells) and a tendency toward enhanced plasmablast frequencies. Although NSM B cells were significantly reduced in our SLE patients, we did not observe a notable decrease in naive B cells, in contrast to prior studies. This could be attributed to the fact that we did not distinguish between naive and transitional B cells (both are CD27^−^IgD^+^), with the latter being reported to be enhanced in lupus settings. 

Apart from altered cell distribution, numerous abnormalities of BCR signaling, which is a master regulator of B cell survival, proliferation, differentiation, and effector functions, have been described in SLE B cells. Among these defects, lupus B cells exhibit decreased expression and impaired activation of the Lyn kinase [30,31], enhanced tyrosine phosphorylation following BCR stimulation [32], and reduced expression of the PTEN (Phosphatase and Tensin Homolog) phosphatase, which leads to enhanced Akt activation [33]. 

BTLA acts as a negative regulator of BCR signaling. Activation of BTLA via HVEM triggers the recruitment of SHP1 (SH2 Domain-Containing Inositol-5-Phosphatase 1/2), thereby decreasing the phosphorylation of key BCR signaling molecules such as Syk (Spleen Tyrosine Kinase), BLNK (B Cell Linker Protein), and PLCγ2 (Phospholipase C Gamma 2) [34]. This, in turn, leads to reduced B cell proliferation, activation, and cytokine secretion [35]. An altered BTLA expression or a defective function of the BTLA pathway may contribute to B-cell mediated diseases. Surprisingly, despite BTLA being highly expressed on peripheral B cells [34], its role on lymphocyte regulation has been mainly analyzed on T cells. Our study aimed to investigate BTLA and HVEM expression in B cell subsets in order to better understand the role of this regulatory pathway in SLE pathogenesis.

Contrary to previously published findings [17], we did not observe a decrease in BTLA expression on naive CD27^+^IgD^−^ B cells. This discrepancy may be attributed to differences in patient selection, particularly regarding medication. In our study, we included only patients who were either untreated or treated with hydroxychloroquine, methotrexate, and/or low doses of steroids. In contrast, 2/3 of the patients in Wiedemann et al.’s study were receiving immunosuppressive drugs such as mycophenolate mofetil, azathioprine, or biologics [17]. However, we confirmed that BTLA expression is significantly reduced on the surface of DN memory B cells from SLE patients compared to HCs, while HVEM remains consistent across all B cell subsets. More importantly, we identified a population of memory B cells with very low levels of surface BTLA in lupus patients. Although BTLA^low^ DN B cells can also be detected in HCs, their frequency is largely enhanced in SLE patients. Previous studies have shown that BTLA expression on B cells declines with age [36], and we similarly observed reduced BTLA expression in SLE patients over 60 years of age. The frequency of DN B cells was also described to be enhanced in elderly individuals [37]. Interestingly, we found a positive correlation between age and the frequency of BTLA^low^ DN B cells in SLE patients, suggesting that diminished BTLA expression could contribute to the higher frequency of this BTLA^low^ cell population in SLE patients. 

In recent years, particular attention has been paid to the DN subset in SLE. In 2018, DN B cells expressing both CD11c and T-bet markers, with the ability to differentiate into antibody-secreting cells, were described in lupus patients [25]. The same year, a second team identified two subpopulations of DN B cells, according to CXCR5, CD11c, and T-bet expression, i.e., DN1 (CXCR5^+^CD11c^−^Tbet^−^) and DN2 (CXCR5^−^CD11c^+^Tbet^+^) B cells [26]. The frequency of DN2 B cells, which are hyper-reactive to TLR7 (Toll-Like Receptor 7) stimulation and are able to produce autoantibodies, is drastically increased in SLE patients. In our study, the frequency of CD11c^+^Tbet^+^ DN2 B cells appears to be elevated in SLE patients, although it does not reach statistical significance. However, given the limitation of the sample size, we can hypothesize that increasing the number of analyzed SLE patients would reveal a statistically significant enhancement of DN2 frequency.

A further characterization of the DN subset has revealed at least four subsets within the DN population, designated DN1 to DN4 B cells, each with distinct expression patterns of CD21, CD11c, CD19 and CXCR5 expression. The DN3 subset was initially described in the context of SARS-CoV2 infection as DN B cells that do not express CD21 nor CD11c [38,39]. DN3 B cell frequency is enhanced in infected patients compared to HCs and appears to play a role in extrafollicular responses. Using scRNAseq and RNA trajectory analysis on DN B cells, Stewart et al. very recently showed that DN subsets can be divided into two distinct developmental branches. They proposed that DN1 and DN4 B cells, expressing CXCR5, follow a T-dependent B cell developmental pathway, whereas T-independent B cell responses occur via a shared pathway involving DN3 and DN2 [40]. 

An in-depth analysis of the BTLA^low^ DN subset that we identified suggests that it mostly corresponds to the DN3 subset. Indeed, BTLA^low^ DN B cells share a phenotypic profile similar to DN3 B cells, i.e., no expression of CXCR5 and low expression of CD19. However, BTLA^low^ DN B cells do not uniformly express CD11c. As previously proposed by others for DN3 B cells, these BTLA^low^ DN B cells exhibit an activated phenotype and share characteristics with antibody-secreting cells. We observed a significant increase in the frequency of DN3 B cells with very low BTLA expression in lupus patients, particularly in those with anti-dsDNA antibodies. Consistent with our findings, Jenks et al. also recently reported that DN3 B cells, along with DN2, are expanded in SLE patients [41]. This highlights the potential role of DN3 B cells in disease immunopathology.

Another group designated three B cell subsets among DN B cells, according to CXCR5 and CD19 expression. They found that CXCR5^−^ DN cells are enhanced in SLE patients compared to HCs, but only the frequency of those expressing low levels of CD19 correlates with plasmablast frequencies and shows higher levels of IRF4 (Interferon Regulatory Factor 4), PRDM1 (PR Domain Zing Finger Protein 1), and XBP1 (X-Box Binding Protein 1) [42]. Notably, these CXCR5^−^CD19^lo^ DN B cells express lower levels of BTLA than their CXCR5^−^CD19^hi^ counterparts, suggesting that the CXCR5^−^CD19^low^ DN cells are likely representative of the DN3 B cell subset, while the CXCR5^−^CD19^hi^ DN B cells correspond to the DN2 subset. Collectively, these data show that DN3 B cells, which express low levels of BTLA and display characteristics of plasmablast precursors, are enriched in lupus patients as supported by both our study and previous findings [42]. 

There is one important question: what role do DN3 B cells play in lupus pathogenesis? The differentiation of autoreactive B cells into autoantibody-secreting cells is a key pathogenic mechanism in SLE. Initially, DN2 B cells were considered the main subset among DN B cells to assume this pathogenic role. However, the DN3 subset may also be significantly involved. In line with this assumption, Chizzolini et al. very recently showed that the DN3 subset is most robustly associated with disease activity than DN2 B cells in lupus patients [43]. Increased frequencies of DN3 B cells have been observed in severe SARS-CoV2 infections [44,45], correlating with inflammatory markers such as CRP (C-Reactive Protein) and proinflammatory cytokines (TNF-α, IL-6, IFN-γ, and IL-1β), as well as elevated levels of serum autoantibodies [45]. This suggests that DN3 B cells could play an important role in the inflammatory processes associated with lupus. Importantly, DN3 B cells are the ones that express the lowest BTLA levels. This reduced BTLA expression could impact the BCR response in SLE patients, potentially counteracting their inhibition. As a result, this may promote the rapid differentiation of DN3 B cells into autoantibody-secreting cells. 

Since DN3 B cells have only recently been described, their ontogeny remains largely unknown. RNA velocity analysis suggests that the DN3 stage may precede the DN2 stage [40]; however, additional studies are required to confirm this hypothesis. Recent evidence indicates that they are not transcriptionally related and exhibit some differences, as contrary to the DN2 B cell subset, DN3 B cells exhibit a strong transcriptional signature of proliferation and the unfolded protein response and are the only DN B cell subset expressing high levels of RNA-encoding IgG4 [27,46]. Our data also underscore the uncertainty surrounding the relationship between the DN2 and DN3 subsets. Indeed, while DN1 B cells are unaffected by low BTLA expression, and the majority of BTLA^low^ DN B cells correspond to DN3, a portion of DN2 cells also fall within this BTLA^low^ population. More research is necessary to determine which originated first, DN2 and DN3 B cells, or whether they represent distinct developmental pathways within the extrafollicular DN B cell population. 

Another potential pathogenic role of DN3 B cells may be linked to their capacity to infiltrate lesions. Interestingly, while BTLA is strongly expressed on peripheral B cells from patients with cardiovascular diseases, effector B cells within lesions, primarily consisting of NSM and DN B cells, express very low levels of BTLA [47]. Additionally, DN3 B cells have been recently reported to infiltrate inflamed tissues in cases of autoimmune fibrosis and severe COVID-19 [27]. Interestingly, DN3 B cells exhibit higher activation levels compared to other DN B cell subsets, as evidenced by CD95 expression, which is twice as high. Consistently, BTLA has been found to be ten-fold downregulated on PMA/ionomycin-activated B cells compared to unstimulated cells [48]. Collectively, these findings suggest that tissue infiltration by activated DN3 cells may contribute to the local inflammatory processes.

Finally, much attention should be paid to DN3 B cells before considering a therapeutic strategy in lupus patients. DN memory B cells have been shown to be significantly reduced in rheumatoid arthritis patients treated with TNF inhibitors and tocilizumab [49], as well as in lupus patients with nephritis undergoing immunosuppressive therapy [50]. While these findings are promising, they did not clarify whether the restoration to normal levels applies to all DN B cell subsets, which warrants further investigations. Indeed, a recent study reported that although rituximab efficiently reduces DN2 frequencies in lupus patients, it leads to a shift toward the DN3 phenotype [51]. Given their relatively low expression of surface molecules compared to other DN B cell subsets, directly targeting DN3 cells may prove challenging. Interestingly, anti-CD19 CAR T-cell therapy was shown to be efficient in SLE patients refractory to several immunosuppressive drugs with remission achieved for all five treated patients [52]. Despite their lower CD19 expression level compared to other DN subsets, one can argue that DN3 B cells were efficiently targeted by anti-CD19 CAR T cell therapy. Nevertheless, alternative strategies could focus on targeting the factors responsible for DN3 generation, and further efforts should be directed toward identifying these factors in the future.

Our study has several shortcomings. First, our data are mainly descriptive, and mechanistic studies would provide functional evidence of the BTLA^low^ DN B cell involvement in lupus pathogenesis. Transcriptomic approaches are needed to better characterize BTLA^low^ DN B cells and their ontogeny, and cellular experiments will help to validate the potential pathogenic function of these cells. Second, we did not find any association between increased BTLA^low^ DN B cell frequency and disease activity and/or severe clinical manifestation such as nephritis. This could be attributed to the limited number of patients with an active disease (8 out of 48 with an active disease and 7 out of 48 with nephritis), and studies in larger cohorts of active SLE patients are thus required to make conclusions. Another limitation is the very few numbers of elderly HCs, preventing us from comparing BTLA expression and DN frequency in older individuals. However, decreased BTLA in SLE DN B cells expression was still evidenced in the age-matched analysis, indicating that the reduced BTLA expression is not only related to age but also to the lupus disease.

## 5. Conclusions

In conclusion, we evidenced a reduced expression of the inhibitory receptor BTLA on DN B cells of SLE patients compared to HCs. Further analysis showed that BTLA expression is drastically reduced in the newly described DN3 B cell subset and to a lower extent in DN2 B cells. While BTLA^low^ DN3 B cells can be detected in HCs, they are significantly enriched in SLE patients. This specific memory B cell subset exhibits an activated phenotype and shares characteristics with antibody-secreting cells. Our data highlight the role of DN3 B cells in SLE pathogenesis and emphasize the importance of considering these cells in the development of future therapeutic strategies. 

## Figures and Tables

**Figure 2 cells-13-02063-f002:**
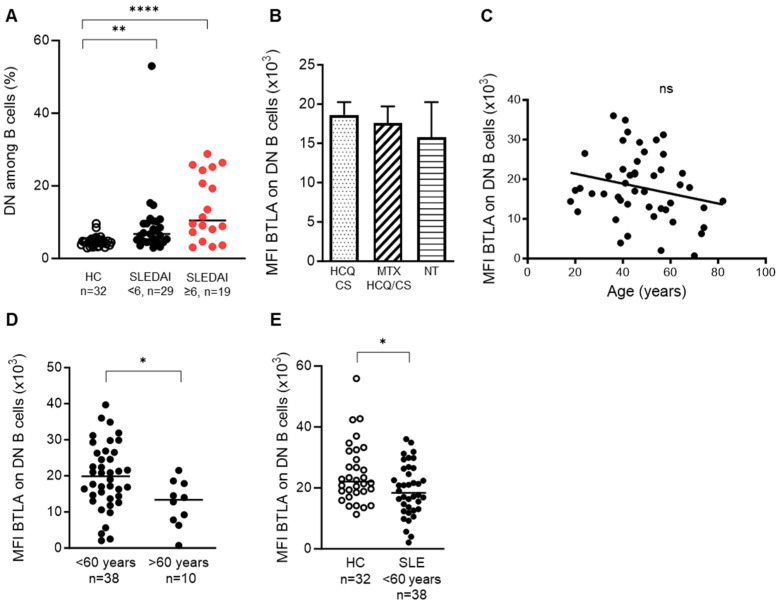
BTLA expression in DN B cells. (**A**) Frequency of DN B cells among CD19^+^ B cells in HCs, SLE patients with inactive or low disease activity (SLEDAI < 6), and patients with mild to severe SLE (SLEDAI ≥ 6). (**B**) BTLA expression (MFI) on DN B cells from SLE patients according to treatments. (**C**) Correlation between BTLA expression on DN B cells from SLE patients and age (*n* = 48). (**D**,**E**) BTLA expression on DN B cells was compared between lupus patients of less than or more than 60 years of age (**D**) and between HC and lupus patients of less than 60 years of age (**E**). Results are expressed as mean ± SEM. HCQ, hydroxychloroquine; CS, corticosteroids; MTX, methotrexate; NT, no treatment. * *p* < 0.05; ** *p* < 0.01; **** *p* < 0.0001, Kruskal–Wallis or Mann–Whitney tests. ns, not significant.

**Figure 3 cells-13-02063-f003:**
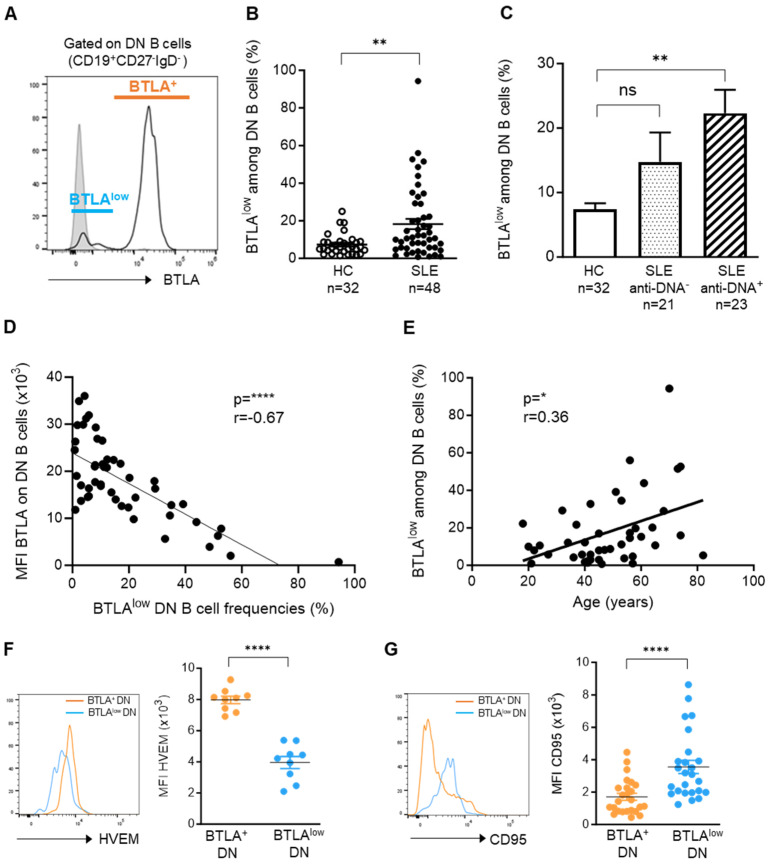
Some DN B cells do not express BTLA. (**A**) Representative histogram of BTLA staining on DN B cells from an SLE patient. (**B**) Frequency of BTLA^low^ DN B cells in HC and SLE patients. (**C**) Frequency of BTLA^low^ DN B cells according to the presence of anti-DNA antibodies. (**D**) Correlation between BTLA expression and BTLA^low^ DN B cell percentages (*n* = 48) in SLE patients. (**E**) Correlation between BTLA^low^ DN B cell frequencies and age (*n* = 44) in SLE patients. (**F**,**G**) Comparison of HVEM (**E**) and CD95 (**F**) expression (MFI) between BTLA^+^ and BTLA^low^ DN B cells in SLE patients. Results are expressed as mean ± SEM. * *p* < 0.05; ** *p* < 0.01; **** *p* < 0.0001, Kruskal–Wallis or Mann–Whitney tests. ns, not significant.

**Figure 4 cells-13-02063-f004:**
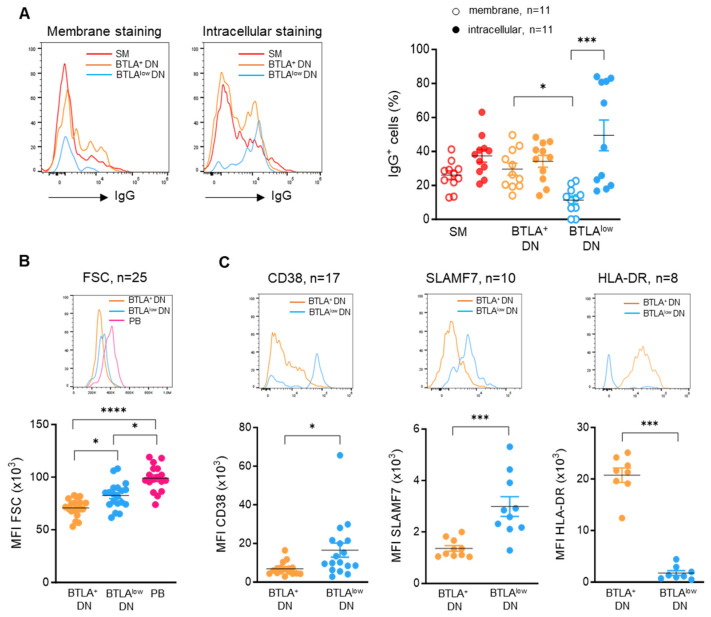
BTLA^low^ DN B cells from SLE patients harbor characteristics of antibody-secreting cells. (**A**) Membrane and intracellular expression of IgG in SM, BTLA^+^, and BTLA^low^ DN B cells. (**B**) Representative image of FSC staining and comparison of FSC values between BTLA^+^, BTLA^low^ DN B cells, and PBs. (**C**) Expression of CD38, SLAMF7, and HLA-DR in BTLA^+^ and BTLA^low^ DN B cells. Results are expressed as mean ± SEM. * *p* < 0.05; *** *p* < 0.001; **** *p* < 0.0001, Kruskal–Wallis or Mann–Whitney tests.

**Figure 5 cells-13-02063-f005:**
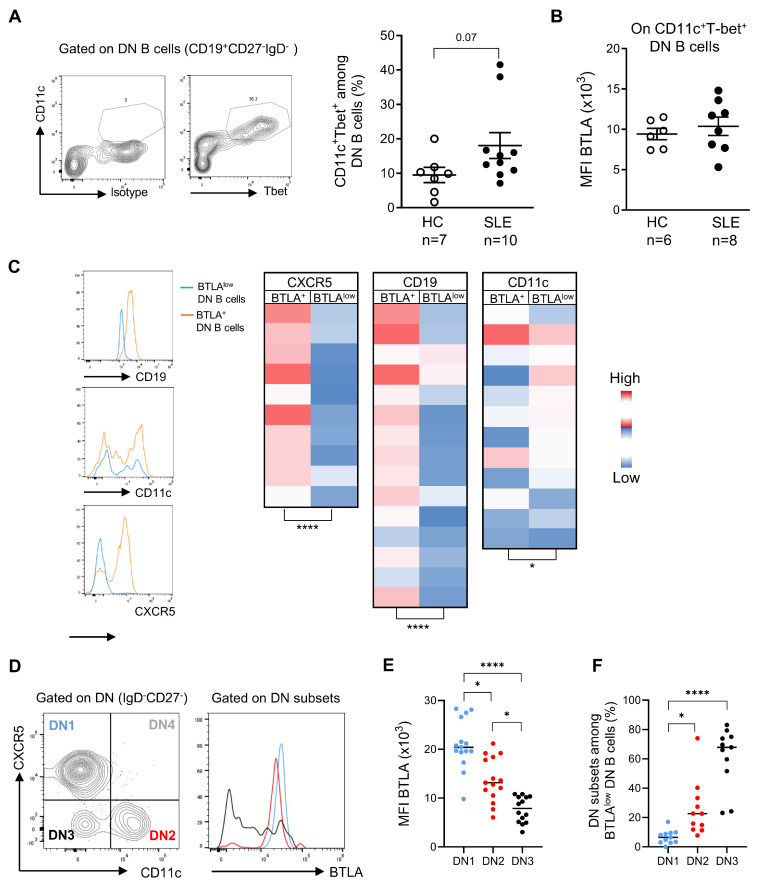
BTLA^low^ DN B cells mostly correspond to the DN3 B cell subset. (**A**) Frequency of atypical memory B cells (CD11c^+^Tbet^+^) among DN memory B cells in HC and SLE patients. (**B**) Comparison of BTLA expression by CD11c^+^Tbet^+^ DN B cells between HC and SLE patients. (**C**) Representative staining (left) and heatmaps of CD19, CD11c, and CXCR5 protein levels in BTLA^+^ and BTLA^low^ DN B cells. Each box corresponds to one patient. (**D**) Gating strategy to discriminate among DN1, DN2, DN3, and DN4 B cell subsets, along with an example of BTLA staining on each subset, are represented. (**E**,**F**) Comparison of BTLA expression in DN1, DN2, and DN3 B cells in SLE patients (*n* = 11). As the DN4 B cell subset was not reproducibly detectable, it was not further analyzed. Results are expressed as mean ± SEM. * *p* < 0.05; **** *p* < 0.0001, Kruskal–Wallis or Mann–Whitney tests.

**Table 1 cells-13-02063-t001:** Clinical and biological characteristics of SLE patients.

	SLE Patients (*n* = 48)
Sex (F/M), *n*	42/6
Age (years), median (range)	45.5 (18–82)
SLEDAI, median (range) In remission (SLEDAI = 0), *n* Low activity (SLEDAI 1–5), *n* Mild activity (SLEDAI 6–10), *n* High activity (SLEDAI 11–19), *n* Very high activity (SLEDAI ≥ 20), *n*	4 (0–23) 13 16 11 4 4
Clinical manifestations *, *n* Arthritis Rash Nephritis Pleurisy/pericarditis	16 9 7 5
Biological features, *n* Anti-dsDNA ** Low complements Proteinuria Hematuria	23 19 11 6
Hematological features, *n* Anemia Lymphopenia Leucopenia Thrombocytopenia	12 11 5 5
Treatment, *n*, median (range) None CS CS ≤ 5 mg/day CS > 5 mg/day HCQ (mg/day) MTX (mg/week)	6 25 6 (2.5–15) 16 3.7 (2.5–5) 9 10 (7–15) 29 400 (200–600) 10 20 (10–25)

F, female; M, male; CS, corticosteroids; HCQ, hydroxychloroquine; MTX, methotrexate. * at the time of blood drawn; ** considered positive when the titer was 50 ≥ IU/mL as measured by ELISA.

## Data Availability

All data that were generated for this study are available upon reasonable request.

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
