# Peer review of "Subset of DN Memory B Cells Expressing Low Levels of Inhibitory Receptor BTLA Is Enriched in SLE Patients"

_cells, 2024, doi:10.3390/cells13242063_

Round 1

Reviewer 1 Report

Comments and Suggestions for Authors

Thank you very much for the opportunity to review this manuscript. Here, Aubergeon and colleagues present a study investigating the expression of BTLA and HVEM on B-cell subsets in lupus. Investigating the potential of BTLA and HVEM as biomarkers and also targets for treatment is important, therefore the manuscript is of interest especially for the rheumatology and nephrology community. The authors have published several author paper regarding BTLA in the past and can be considered experts in the field.

The authors report the identification of a subset of B-cells, which is low in BTLA expression and has an antibody secreting phenotype. They conclude that these cell might be important for disease activity in lupus.

While the research topic is important and the finding in general is highly interesting, in general the data presentation should be improved and is in this state not convincing for all experiments. The authors should be encouraged to conduct further studies to understand the functionality of the identified B-Cell subsets, which would make the study more informative and the conclusion more convincing.

I have some suggestions, which might be useful for improvement of the manuscript.

Figure 1:

-              The decreased expression of BTLA on DN B-cells is a central observation in the manuscript. However, the changes are rather subtle and can moreover not easily be appreciated in the corresponding histogram. I strongly suggest showing the histograms for the individual cell populations separately alongside with the respective controls. It would be very informative if the authors show also regulation on the transcriptomic (e.g. qPCR or even Bulk RNAseq) vs. surface protein expression level. This would corroborate their finding and might conclusions regarding transcriptomic or (post)translational regulation possible.

-              As a general comment: Throughout the manuscript, many figures miss labeling on the X-or Y-axis (scale) and a clear statement of the pregating in the respective experiment. This makes the interpretation of the data quality very hard. Whenever possible representative dot-plots or histograms have to be shown. If there is limited space, this could be moved to the supplement.

Figure 2:

-              In A and D representative dot plots/ histograms should be shown

Figure 3:

-              For each experiment the pregating strategy should be clearly stated. Representative FACS plots should be shown to make the data representation more convincing.

Figure 4:

-              The difference in Tbet expression is not statistically significant, therefore the statement that “the enhancement of Tbet positive cells” was “confirmed” should not be made.

-              The differential expression in the heatmap should be statistically tested, it is not possible to judge whether this is a statistically significant effect. As stated also above, using a different analysis technique, e.g. qPCR or even bulk RNAsequencing might yield more convincing results and might be instrumental in painting a bigger picture regarding the function of this particular subset.

-              4d: again proper labeling of the scale of the axes in the representative plots are missing

-              Further mechanistic studies are needed to make assumptions regarding the functionality of the BTLA lacking B-cells.

Discussion:

-              The data of the study are mostly correlations. Limitations of the study should be discussed more thoroughly in the discussion, these include in the opinion of the reviewer, the lack of mechanistic studie, e.g. in mouse/ cell culture models to show causality between the reduced expression of BTLA on the DN B-cells and enhanced B-cell activity, better characterization of the secretory phenotype and in general addition of qPCR/ transcriptomic studies in the future to better understand the functionality of BTLAlow B-cells.

Author Response

Thank you for taking the time to review our manuscript and providing us with valuable feedback. The detailed answers to your remarks are given below in a point-by-point fashion

Figure 1:

-              The decreased expression of BTLA on DN B-cells is a central observation in the manuscript. However, the changes are rather subtle and can moreover not easily be appreciated in the corresponding histogram. I strongly suggest showing the histograms for the individual cell populations separately alongside with the respective controls.

The figure was changed accordingly to the reviewer’s request and a supplementary Figure S1 showing the gating strategy was added. The comparison of BTLA and HVEM expression between B cell subsets (both in HC and SLE patients) is now depicted in supplementary Figure S2.

 It would be very informative if the authors show also regulation on the transcriptomic (e.g. qPCR or even Bulk RNAseq) vs. surface protein expression level. This would corroborate their finding and might conclusions regarding transcriptomic or (post)translational regulation possible.

We agree with the reviewer that such analyses would add subsequent information. However, performing such experiments would require the recruitment of new SLE patients. SLE being a rare disease, it will take a long time, and more importantly, the number of BTLAlow DN B cells (even in SLE patients) is relatively low and thus limits the possibility of conducting such in vitro cell culture experiments to evaluate their function. We added a paragraph on this limitation in the discussion as suggested by another reviewer (lines 414 to 426).

-              As a general comment: Throughout the manuscript, many figures miss labeling on the X-or Y-axis (scale) and a clear statement of the pregating in the respective experiment.This makes the interpretation of the data quality very hard. Whenever possible representative dot-plots or histograms have to be shown. If there is limited space, this could be moved to the supplement.

This was corrected and this information was added to the new figures.

Figure 2:

-              In A and D representative dot plots/ histograms should be shown.

These representative dot plots showing DN frequencies in HC and SLE patients are already depicted in Figure 1A and the histogram representative for BTLA expression is shown in Figure 1C.

Figure 3:

-              For each experiment the pregating strategy should be clearly stated. Representative FACS plots should be shown to make the data representation more convincing.

Pregating strategy is always the same and is depicted in Figure 1. The statement of pregating strategy was added in figures.

Figure 4: (we suppose that the reviewer means Figure 5)

-              The difference in Tbet expression is not statistically significant, therefore the statement that “the enhancement of Tbet positive cells” was “confirmed” should not be made.

The sentence in the result section was modified accordingly (lines 243-245) and this point was added to the discussion (lines 324-328).

-              The differential expression in the heatmap should be statistically tested, it is not possible to judge whether this is a statistically significant effect.

Statistical analyses were performed, we apologize for this missing information. Results are now shown in the figure.

-              4d: again proper labeling of the scale of the axes in the representative plots are missing.

This was corrected accordingly to the reviewer’s request.

-              Further mechanistic studies are needed to make assumptions regarding the functionality of the BTLA lacking B-cells.

A paragraph about the limitations of our study was added in the discussion (lines 414-426).

Discussion:

-              The data of the study are mostly correlations. Limitations of the study should be discussed more thoroughly in the discussion, these include in the opinion of the reviewer, the lack of mechanistic studie, e.g. in mouse/ cell culture models to show causality between the reduced expression of BTLA on the DN B-cells and enhanced B-cell activity, better characterization of the secretory phenotype and in general addition of qPCR/ transcriptomic studies in the future to better understand the functionality of BTLAlow B-cells.

A paragraph about the limitations of our study was added in the discussion (lines 414-426).

Reviewer 2 Report

Comments and Suggestions for Authors

Comment 1: Lots of abbreviations are used in the text, so it is suggested that an abbreviation list can be added.

Comment 2: Not all abbreviations used in the manuscript are explained in the text (for example, BCR and TCR line 33, MRI line 46, BCR line 257, and much more in the discussion section)

Comment 3: HVEM is explained twice in the text, once in capital letters and once in small letters.

Comment 4: A more detailed description of the interaction of BTLA and HVEM would be useful in the introduction. Especially since the discussion suddenly includes a description of the consequences of this interaction (lines 263-267). Maybe a figure that would familiarize the reader with the interaction of these two proteins??

Comment 5: Table 1 is not clear to the reviewer. Why is the number n in the individual parameters not equal to 48?? I understand it can be larger (one patient can have, e.g., several biological features), but why is it sometimes smaller?? This means that the characteristic does not apply to all patients.

Comment 6: The formatting in Table 1 (in the treatment section) needs to be corrected.

Comment 7: It is suggested that line spacing be unified in manuscripts. In the statistic analysis section, it is  different from the rest of the text.

Comments on the Quality of English Language

Manuscript can benefit from language editiong.

Author Response

Thank you for taking the time to review our manuscript and providing us with valuable feedback. The detailed answers to your remarks are given below in a point-by-point fashion

Comment 1Lots of abbreviations are used in the text, so it is suggested that an abbreviation list can be added and Comment 2Not all abbreviations used in the manuscript are explained in the text (for example, BCR and TCR line 33, MRI line 46, BCR line 257, and much more in the discussion section)

Abbreviations are now explained in the text, except for MRL which is classically used and because indicating its abbreviation would have made the text cumbersome.

For the reviewer information, MRL stands for “Murphy Roths Large » and lpr for  « lymphoproliferation »

Comment 3HVEM is explained twice in the text, once in capital letters and once in small letters.

This was corrected (in capital letters)

Comment 4: A more detailed description of the interaction of BTLA and HVEM would be useful in the introduction. Especially since the discussion suddenly includes a description of the consequences of this interaction (lines 263-267). Maybe a figure that would familiarize the reader with the interaction of these two proteins??

A paragraph was added in the introduction (lines 32-40)

Comment 5: Table 1 is not clear to the reviewer. Why is the number n in the individual parameters not equal to 48?? I understand it can be larger (one patient can have, e.g., several biological features), but why is it sometimes smaller?? This means that the characteristic does not apply to all patients.

This is due to the fact that some patients only have biological features such as anti-dsDNA antibodies without any clinical manifestation or hematological feature.

Comment 6: The formatting in Table 1 (in the treatment section) needs to be corrected.

This was done accordingly

Comment 7: It is suggested that line spacing be unified in manuscripts. In the statistic analysis section, it is  different from the rest of the text.

This was corrected.

Reviewer 3 Report

Comments and Suggestions for Authors

This article describes the expression of the inhibitory receptor BTLA and its ligand HVEM on different B cell subsets in SLE patients compared to HC. This is a relevant topic, as BTLA is a negative regulator of B cell receptor signaling, leading to reduced B cell proliferation, differentiation, survival, and function. Consequently, it could contribute to the altered B cell distribution extensively described in SLE. In addition, studies in a lupus mouse model have shown that genetic deletion of BTLA leads to worsened disease. However, previous research on BTLA in the context of autoimmunity has mainly focused on T cells, whereas BTLA expression on B cells remains underexplored. Although this paper is mainly descriptive rather than investigating underlying molecular mechanisms, it provides further evidence that DN B cells are increased and activated in SLE and could contribute to SLE pathogenesis.

Feedback:

1. Since age can influence the frequency of DN B cells as well as BTLA expression, it remains unclear whether the reduced BTLA expression observed in DN B cells from SLE patients compared to HC is due to the disease itself or age differences between SLE patients and HC. Therefore, the authors should specify if the HC are age-matched and if not, it is recommended to provide the age range of the HC and discuss whether age may influence the results presented throughout the paper. In addition, authors should specify how many SLE patients and HC are > 60 years

Materials and methods

2. Line 106: FF-W versus FS-W plots: I assume this is a typing error. Please adapt.

3. The authors should indicate the volume or dilution of the antibodies used for flow cytometry to ensure the reproducibility of the methods

Results

4. Line 131-132: It is recommended to rephrase this sentence as, according to the graph in Figure 1C, the significantly increased expression of BTLA is only seen in naive B cells from HC compared to SM and DN B cells, while there is no significant difference in SLE patients.

5. Line 137-138: ‘HVEM expression was found to be significantly lower in SM B cells compared to NSM B cells’: specify that this is for HC

6. Figure 1A: the authors should indicate in the caption if the HC and SLE patient are age-matched

7. Line 147: add HVEM; ‘BTLA or HVEM expression’

8. Line 148: It is unclear how many patients were screened in each panel in Figure 1

9. Line 155-158 (Figure 2B): the authors should consider tempering this result and the title of Figure 2, as the correlation coefficient is quite low, and the significance may be due to the outlier.

10. Line 161-162: The authors showed a reduced BTLA expression in SLE patients > 60y. Authors should also indicate if this is also the case for aged HC

11. Figure 2D (caption): the authors should indicate the number of SLE patients in each group

12. In section 3.1, the authors state that BTLA is present at the surface of all B cells, but in section 3.3, it appears that DN3 cells are BTLAlow. What about the other B cell subsets? Are there also BTLAlow B cells in the other subsets or is this specific for DN3 cells?

13. Figure 3C: The number of HC should be mentioned as well

14. Figure 3D: The correlation coefficient is not shown on the graph or mentioned in the text

15. Figure 3D-F: the authors should specify in the text and figure caption whether the BTLAlow DN B cells shown are from SLE patients or HC

16. Figure 4: The authors should specify in the caption and in the text of section 3.3 that this analysis is on DN B cells from SLE patients. Data on HC is missing in Figure 4, raising the question whether BTLAlow DN B cells from HC also have characteristics of antibody-secreting cells.

17. Figure 5A-B: the sample size appears to be too small, and the trend towards an increased frequency of CD11c+T-bet+ DN B cells would likely reach significance with a larger sample size. It is recommended that the authors address this point in the discussion section of the article.

18. According to Figure 5E, DN3 cells have the lowest BTLA expression. However, it is not entirely clear whether the BTLAlow DN B cells correspond exclusively to the DN3 subset, as Figure 5C shows some CD11c expression in the BTLAlow DN B cells. It is recommended to check the frequency of the DN subsets within the BTLAlow DN B cells to confirm that they are DN3 cells.

19. Figure 5E: is this also the case for DN3 cells of HC or is there a difference between DN B cells from HC and SLE patients?

Discussion

20. The authors should consider discussing the limitations of this study (e.g. only descriptive with no underlying mechanisms investigated through in vitro functional assays, small sample size for DN subset analysis) as well as future experiments that could further elucidate the underlying mechanism of the BTLA pathway

21. Line 312-314: A recently published article (DOI: 10.1016/j.jtauto.2024.100252) reports that DN3 cells are associated with SLE disease activity. The authors could consider whether referencing this publication could add value to the discussion by highlighting the potential importance of DN3 cells in SLE pathogenesis.  

22. Line 340-341: two times ‘Indeed’. Please rephrase 

Author Response

Thank you for taking the time to review our manuscript and providing us with valuable feedback. The detailed answers to your remarks are given below in a point-by-point fashion

Feedback:

Since age can influence the frequency of DN B cells as well as BTLA expression, it remains unclear whether the reduced BTLA expression observed in DN B cells from SLE patients compared to HC is due to the disease itself or age differences between SLE patients and HC. Therefore, the authors should specify if the HC are age-matched and if not, it is recommended to provide the age range of the HC and discuss whether age may influence the results presented throughout the paper. In addition, authors should specify how many SLE patients and HC are > 60 years.

We thank the reviewer for his very interesting suggestion, and these analyses were added in the revised version of our manuscript.

The age of HC was added in the M&M section. As HC are not perfectly age-matched with SLE patients (range from 21 to 63 and median of 41.2), and because only 2 HC were >60 year-old, we also compared BTLA expression in DN B cells when SLE patients that are older than 60 years were excluded from the analysis and still found a significant decreased BTLA expression on DN from SLE patients (Figure 2E).  This led us to conclude that even if BTLA expression is decreased with age, the reduced expression in SLE DN B cells is also related to the disease itself.

Materials and methods

  1. Line 106: FF-W versus FS-W plots: I assume this is a typing error. Please adapt. This was indeed a typing error and this was corrected.
  2. The authors should indicate the volume or dilution of the antibodies used for flow cytometry to ensure the reproducibility of the methods: Dilution were added for all antibodies used in this study.

Results

  1. Line 131-132: It is recommended to rephrase this sentence as, according to the graph in Figure 1C, the significantly increased expression of BTLA is only seen in naive B cells from HC compared to SM and DN B cells, while there is no significant difference in SLE patients. In fact, there is also significant differences in SLE. As requested by another reviewer, this figure was changed to show differential BTLA (and HVEM) expression on each B cell subset separately. The comparison of BTLA and HVEM expression on B cell subsets was moved to supplementary Figure S2, and was depicted for both HC and SLE patients.
  2. Line 137-138: ‘HVEM expression was found to be significantly lower in SM B cells compared to NSM B cells’: specify that this is for HC. This has been specified.
  3. Figure 1A: the authors should indicate in the caption if the HC and SLE patient are age-matched this point is discussed with figure 2. We added the term “partially age-matched” line 138.
  4. Line 147: add HVEM; ‘BTLA or HVEM expression’ This was added
  5. Line 148: It is unclear how many patients were screened in each panel in Figure 1. The number of HC and SLE patients was added in each panel.
  6. Line 155-158 (Figure 2B): the authors should consider tempering this result and the title of Figure 2, as the correlation coefficient is quite low, and the significance may be due to the outlier.

We agree with the reviewer and finally, we decided to delete the sentence and the corresponding figure. Instead, we added in figure 2, the graph showing correlation between BTLA expression on DN B cells and age (Figure 2C) and as it is more relevant, a new panel in Figure 3 (Figure 3D) showing the inverse correlation of BTLA expression and the enhancement of BTLAlow DN B cell frequencies. This result is described in the result section (lines 191-194).

  1. Line 161-162: The authors showed a reduced BTLA expression in SLE patients > 60y. Authors should also indicate if this is also the case for aged HC. As we only have two HC older than 60 years it was not possible to perform such analysis.
  2. Figure 2D (caption): the authors should indicate the number of SLE patients in each group; This was added.
  3. In section 3.1, the authors state that BTLA is present at the surface of all B cells, but in section 3.3, it appears that DN3 cells are BTLAlow. What about the other B cell subsets? Are there also BTLAlowB cells in the other subsets or is this specific for DN3 cells? BTLA low expression was only observed in DN B cells. We added “(but not other B cell subsets)” in the sentence (line 187)
  4. Figure 3C: The number of HC should be mentioned as well: This was added
  5. Figure 3D: The correlation coefficient is not shown on the graph or mentioned in the text: This was added on the figure.
  6. Figure 3D-F: the authors should specify in the text and figure caption whether the BTLAlowDN B cells shown are from SLE patients or HC. This was done.
  7. Figure 4: The authors should specify in the caption and in the text of section 3.3 that this analysis is on DN B cells from SLE patients. Data on HC is missing in Figure 4, raising the question whether BTLAlowDN B cells from HC also have characteristics of antibody-secreting cells. We specified that data are from SLE patients in Figure 4, and we added characteristics of BTLAlow DN B cells from HC in supplementary Figure S3. A sentence was added concerning this B cell subset from HC in the result section (lines 234-236).
  8. Figure 5A-B: the sample size appears to be too small, and the trend towards an increased frequency of CD11c+T-bet+ DN B cells would likely reach significance with a larger sample size. It is recommended that the authors address this point in the discussion section of the article. The sentence in the Result section was modified (lines 243-245) and this point was added in the discussion (lines 324-328).
  9. According to Figure 5E, DN3 cells have the lowest BTLA expression. However, it is not entirely clear whether the BTLAlowDN B cells correspond exclusively to the DN3 subset, as Figure 5C shows some CD11c expression in the BTLAlow DN B cells. It is recommended to check the frequency of the DN subsets within the BTLAlow DN B cells to confirm that they are DN3 cells.

We completely agree with the reviewer. CD11c is not uniformly expressed BTLAlow DN B cells, with cells expressing CD11c and others that do not express CD11c. Consequently, DN2 B cells also express lower levels of BTLA than DN1 B cells (Figure 5). As suggested by the reviewer, we analyzed the frequency of each DN subsets among BTLAlow DN B cells and as expected, found that even if the majority of BTLAlow DN B cells are DN3 B cells, a small but not negligible fraction are DN2 B cells. This analysis was added in Figure 5 (as Figure 5F) and we modified accordingly our conclusion “ BTLAlow DN B cells mainly corresponding to DN3 B cells”. We also added 2 sentences in the discussion section (line 380-383).

  1. Figure 5E: is this also the case for DN3 cells of HC or is there a difference between DN B cells from HC and SLE patients? It is also the case for HC and this analysis was added in the revised manuscript (line 267 and supplementary Figure 4)

Discussion

  1. The authors should consider discussing the limitations of this study (e.g. only descriptive with no underlying mechanisms investigated through in vitro functional assays, small sample size for DN subset analysis) as well as future experiments that could further elucidate the underlying mechanism of the BTLA pathway. A paragraph concerning limitations of our study was added (lines 414-426).
  2. Line 312-314: A recently published article (DOI: 10.1016/j.jtauto.2024.100252) reports that DN3 cells are associated with SLE disease activity. The authors could consider whether referencing this publication could add value to the discussion by highlighting the potential importance of DN3 cells in SLE pathogenesis.  We thank the reviewer for this suggestion, which effectively highlights the potential role of DN3 B cells in lupus pathogenesis. This reference was added (lines 364-366).

  1. Line 340-341: two times ‘Indeed’. Please rephrase. This was done accordingly.

Round 2

Reviewer 1 Report

Comments and Suggestions for Authors

The authors have adequately adressed the issues.

Author Response

We thank the reviewer for taking the time to review our manuscript.

Reviewer 3 Report

Comments and Suggestions for Authors

I would like to thank the authors for addressing my comments in their revised manuscript. The manuscript has been noticeably improved. However, I have a few additional remarks (indicated in red) regarding the revisions.

4. Line 131-132: It is recommended to rephrase this sentence as, according to the graph in Figure 1C, the significantly increased expression of BTLA is only seen in naive B cells from HC compared to SM and DN B cells, while there is no significant difference in SLE patients. In fact, there is also significant differences in SLE. As requested by another reviewer, this figure was changed to show differential BTLA (and HVEM) expression on each B cell subset separately. The comparison of BTLA and HVEM expression on B cell subsets was moved to supplementary Figure S2, and was depicted for both HC and SLE patients. I appreciate the new figures, however, it is still recommended to rephrase this sentence (lines 144-145 in the revised version) as significantly increased BTLA expression has only been shown for naive B cells and not for NSM B cells. If it is also significant for NSM, this should be indicated in the graphs in Figure S2.

8. Line 148: It is unclear how many patients were screened in each panel in Figure 1. The number of HC and SLE patients was added in each panel. I would like to thank the authors for adding the number of patients to the graphs in Figure 1. However, it appears unusual that the number of HC and SLE patients is lower for PB compared to the other B cell subsets. This suggests that the authors may have used different flow cytometry panels. If this is the case, BTLA expression in the different B cell subsets should only be compared using donors for which data is available for all 5 subsets. The authors should clarify the differences in donor numbers. 

9. Line 155-158 (Figure 2B): the authors should consider tempering this result and the title of Figure 2, as the correlation coefficient is quite low, and the significance may be due to the outlier. We agree with the reviewer and finally, we decided to delete the sentence and the corresponding figure. Instead, we added in figure 2, the graph showing correlation between BTLA expression on DN B cells and age (Figure 2C) and as it is more relevant, a new panel in Figure 3 (Figure 3D) showing the inverse correlation of BTLA expression and the enhancement of BTLAlow DN B cell frequencies. This result is described in the result section (lines 191-194). I would like to thank the authors for agreeing with my comment. However, in Figure 3D, do the authors show the correlation between BTLA expression on DN B cells and % BTLAlow B cells or DN B cells? The X-axis mentions ‘B cells’, but in the caption refers to ‘DN B cells’. In addition, I do not completely understand what the authors aim to demonstrate with Figure 3D. If BTLA expression in the total DN B cell population is lower, would it not logically follow that there are also more BTLAlow DN B cells? Or am I misunderstanding this?

For Figures 3F and G, BTLAlow DN B cells are indicated in orange on the histograms but in blue on the corresponding graphs. Could it be possible that the authors accidentally switched the colors on the histograms? I am concerned this may be the case for all histograms throughout the manuscript and supplementary figures.

16. Figure 4: The authors should specify in the caption and in the text of section 3.3 that this analysis is on DN B cells from SLE patients. Data on HC is missing in Figure 4, raising the question whether BTLAlowDN B cells from HC also have characteristics of antibody-secreting cells. We specified that data are from SLE patients in Figure 4, and we added characteristics of BTLAlow DN B cells from HC in supplementary Figure S3. A sentence was added concerning this B cell subset from HC in the result section (lines 234-236). Line 234-236: ‘not performed in all HC samples’: the authors should indicate the number of HC samples that were used.

18. According to Figure 5E, DN3 cells have the lowest BTLA expression. However, it is not entirely clear whether the BTLAlowDN B cells correspond exclusively to the DN3 subset, as Figure 5C shows some CD11c expression in the BTLAlow DN B cells. It is recommended to check the frequency of the DN subsets within the BTLAlow DN B cells to confirm that they are DN3 cells. We completely agree with the reviewer. CD11c is not uniformly expressed BTLAlow DN B cells, with cells expressing CD11c and others that do not express CD11c. Consequently, DN2 B cells also express lower levels of BTLA than DN1 B cells (Figure 5). As suggested by the reviewer, we analyzed the frequency of each DN subsets among BTLAlow DN B cells and as expected, found that even if the majority of BTLAlow DN B cells are DN3 B cells, a small but not negligible fraction are DN2 B cells. This analysis was added in Figure 5 (as Figure 5F) and we modified accordingly our conclusion “ BTLAlow DN B cells mainly corresponding to DN3 B cells”. We also added 2 sentences in the discussion section (line 380-383). I would like to thank the authors for performing the extra analysis. Based on these results and the changes in the manuscript, it is recommended to attenuate the sentence in line 429-430 from the revised manuscript.

Author Response

I would like to thank the authors for addressing my comments in their revised manuscript. The manuscript has been noticeably improved. However, I have a few additional remarks (indicated in red) regarding the revisions.

We thank the reviewer for carefully reviewing the revised version of our manuscript and providing us insightful comment. Point-by-point response is provided below (in green)

  1. Line 131-132: It is recommended to rephrase this sentence as, according to the graph in Figure 1C, the significantly increased expression of BTLA is only seen in naive B cells from HC compared to SM and DN B cells, while there is no significant difference in SLE patients. In fact, there is also significant differences in SLE. As requested by another reviewer, this figure was changed to show differential BTLA (and HVEM) expression on each B cell subset separately. The comparison of BTLA and HVEM expression on B cell subsets was moved to supplementary Figure S2, and was depicted for both HC and SLE patients. I appreciate the new figures, however, it is still recommended to rephrase this sentence (lines 144-145 in the revised version) as significantly increased BTLA expression has only been shown for naive B cells and not for NSM B cells. If it is also significant for NSM, this should be indicated in the graphs in Figure S2.

We thank the reviewer, we effectively forgot to modify the text. It was corrected (lines 144-145)

  1. Line 148: It is unclear how many patients were screened in each panel in Figure 1. The number of HC and SLE patients was added in each panel. I would like to thank the authors for adding the number of patients to the graphs in Figure 1. However, it appears unusual that the number of HC and SLE patients is lower for PB compared to the other B cell subsets. This suggests that the authors may have used different flow cytometry panels. If this is the case, BTLA expression in the different B cell subsets should only be compared using donors for which data is available for all 5 subsets. The authors should clarify the differences in donor numbers. 

The same panel was used for flow cytometry analysis, but we did not analyze the CD38 marker when we started the project. This is why we were not able to analyze PB in samples that were collected at the beginning of the project.

  1. Line 155-158 (Figure 2B): the authors should consider tempering this result and the title of Figure 2, as the correlation coefficient is quite low, and the significance may be due to the outlier. We agree with the reviewer and finally, we decided to delete the sentence and the corresponding figure. Instead, we added in figure 2, the graph showing correlation between BTLA expression on DN B cells and age (Figure 2C) and as it is more relevant, a new panel in Figure 3 (Figure 3D) showing the inverse correlation of BTLA expression and the enhancement of BTLAlowDN B cell frequencies. This result is described in the result section (lines 191-194). I would like to thank the authors for agreeing with my comment. However, in Figure 3D, do the authors show the correlation between BTLA expression on DN B cells and % BTLAlow B cells or DN B cells? The X-axis mentions ‘B cells’, but in the caption refers to ‘DN B cells’.

It was mistake and the term “ DN” was missing. It was corrected in the axis.

In addition, I do not completely understand what the authors aim to demonstrate with Figure 3D. If BTLA expression in the total DN B cell population is lower, would it not logically follow that there are also more BTLAlow DN B cells? Or am I misunderstanding this?

It was not so obvious, and we indeed observed some patients for which BTLA expression was reduced on DN B cells (6 with MFI below 20x103) but who still have low BTLAlow DN B cell frequencies (less than 10%). But we agree with the reviewer that even if it should be demonstrated, this result is not surprising and we added the term “logically” in the sentence (line 192).

For Figures 3F and G, BTLAlow DN B cells are indicated in orange on the histograms but in blue on the corresponding graphs. Could it be possible that the authors accidentally switched the colors on the histograms? I am concerned this may be the case for all histograms throughout the manuscript and supplementary figures.

It is effectively an error. We apologize and the colors in histograms were modified

  1. Figure 4: The authors should specify in the caption and in the text of section 3.3 that this analysis is on DN B cells from SLE patients. Data on HC is missing in Figure 4, raising the question whether BTLAlowDN B cells from HC also have characteristics of antibody-secreting cells. We specified that data are from SLE patients in Figure 4, and we added characteristics of BTLAlowDN B cells from HC in supplementary Figure S3. A sentence was added concerning this B cell subset from HC in the result section (lines 234-236). Line 234-236: ‘not performed in all HC samples’: the authors should indicate the number of HC samples that were used. It was done to 10 HC for all parameters, but due to the low number of DN B cells in HC, it was sometimes impossible to objectively analyze some staining. We decided to show in the supplementary Figure S3, those for which we managed to have at least 3 HC samples. It was clarified in the text (lines 234-235).
  2. According to Figure 5E, DN3 cells have the lowest BTLA expression. However, it is not entirely clear whether the BTLAlowDN B cells correspond exclusively to the DN3 subset, as Figure 5C shows some CD11c expression in the BTLAlowDN B cells. It is recommended to check the frequency of the DN subsets within the BTLAlowDN B cells to confirm that they are DN3 cells. We completely agree with the reviewer. CD11c is not uniformly expressed BTLAlow DN B cells, with cells expressing CD11c and others that do not express CD11c. Consequently, DN2 B cells also express lower levels of BTLA than DN1 B cells (Figure 5). As suggested by the reviewer, we analyzed the frequency of each DN subsets among BTLAlow DN B cells and as expected, found that even if the majority of BTLAlow DN B cells are DN3 B cells, a small but not negligible fraction are DN2 B cells. This analysis was added in Figure 5 (as Figure 5F) and we modified accordingly our conclusion “ BTLAlow DN B cells mainly corresponding to DN3 B cells”. We also added 2 sentences in the discussion section (line 380-383). I would like to thank the authors for performing the extra analysis. Based on these results and the changes in the manuscript, it is recommended to attenuate the sentence in line 429-430 from the revised manuscript.

It was done accordingly (lines 430-432).